# Changing to a Low-Polyphenol Diet Alters Vascular Biomarkers in Healthy Men after Only Two Weeks

**DOI:** 10.3390/nu10111766

**Published:** 2018-11-14

**Authors:** Sara Hurtado-Barroso, Paola Quifer-Rada, José Fernando Rinaldi de Alvarenga, Silvia Pérez-Fernández, Anna Tresserra-Rimbau, Rosa M. Lamuela-Raventos

**Affiliations:** 1Department of Nutrition, Food Sciences and Gastronomy, School of Pharmacy and Food Sciences, University of Barcelona, 08028 Barcelona, Spain; sara.hurtado_17@ub.edu (S.H.-B.); pquifera@gmail.com (P.Q-R.); zehfernando@gmail.com (J.F.R.d.A.); 2CIBER Physiopathology of Obesity and Nutrition (CIBEROBN), Institute of Health Carlos III, 28029 Madrid, Spain; anna.tresserra@iispv.cat; 3INSA-UB, Nutrition and Food Safety Research Institute, University of Barcelona, 08028 Barcelona, Spain; 4Department of Endocrinology & Nutrition, CIBER of Diabetes and Associated Metabolic Diseases, Biomedical Reseacrh Institute Sant Pau, Hospital de la Santa Creu i Sant Pau, 08041 Barcelona, Spain; 5Cardiovascular Epidemiology and Genetics Group, REGICOR, IMIM (Instituto Hospital del Mar de Investigaciones Médicas), 08003 Barcelona, Spain; sperez@imim.es; 6Center for the Biomedical Research Network in Cardiovascular Diseases (CIBERCV), Institute of Health Carlos III, 28029 Madrid, Spain; 7Human Nutrition Unit, University Hospital of Sant Joan de Reus, Department of Biochemistry and Biotechnology, Faculty of Medicine and Health Sciences, Pere Virgili Health Research Center, University Rovira i Virgili, 43201 Reus, Spain

**Keywords:** bioactive compounds, nitric oxide, vascular biomarkers, low antioxidant diet, Mediterranean diet, eicosanoids, adhesion molecules

## Abstract

Bioactive dietary compounds play a critical role in health maintenance. The relation between bioactive compound intake and cardiovascular health-related biomarkers has been demonstrated in several studies, although mainly with participants who have altered biochemical parameters (high blood pressure, high cholesterol, metabolic syndrome, etc.). The aim of this study was to evaluate if adopting a diet low in polyphenol-rich food for two weeks would affect vascular biomarkers in healthy men. In a crossover study, 22 healthy men were randomly assigned to their usual diet (UD), consuming healthy food rich in polyphenols, or to a low antioxidant diet (LAD), with less than two servings of fruit and vegetables per day and avoiding the intake of cocoa products, coffee and tea. As a marker of compliance, total polyphenols in urine were significantly lower after the LAD than after the UD (79 ± 43 vs. 123 ± 58 mg GAE/g creatinine). Nitric oxide levels were also reduced (52 ± 28 in LAD vs. 80 ± 34 µM in UD), although no significant changes in cellular adhesion molecules and eicosanoids were observed; however, an increasing ratio between thromboxane A_2_ (TXA_2_) and prostaglandin I_2_ (PGI_2_) was reached (*p* = 0.048). Thus, a slight dietary modification, reducing the consumption of polyphenol-rich food, may affect vascular biomarkers even in healthy individuals.

## 1. Introduction

It is broadly accepted that bioactive compounds of plant origin play a crucial protective role against non-communicable diseases (NCDs). Polyphenols are plant secondary metabolites with one or more aromatic rings that bear at least one hydroxyl group. More than 8000 phenolic structures have been described [1] and several hundred are found in edible vegetables [2]. In the last years, phenolic compounds have been widely studied due to their close association with the prevention of NCDs such as cardiovascular diseases (CVDs) [3,4,5]. Polyphenols are known to have anti-inflammatory properties and improve endothelial function [6,7,8].

Current research about the protective role of polyphenols against CVD is focused on their action on the endothelium [9,10,11]. Endothelium-dependent vasodilator factors include molecules like nitric oxide (NO) and prostacyclin (PGI_2_), while the endothelium-dependent vasoconstrictor response involves reactive oxygen species (ROS) and thromboxane A_2_ (TXA_2_) [12]. NO is a gaseous signaling molecule mainly produced from the amino acid L-arginine by nitric oxide synthase [13] with a crucial role in the prevention of CVD and other NCDs. Endothelium-derived NO is a powerful vasodilator that inhibits platelet adhesion and aggregation as well as thrombin formation, thus preventing vascular injuries. Moreover, NO can inhibit the expression of vascular cell and intercellular adhesion molecules (VCAM-1 and ICAM-1) [14,15].

The entry of polyphenols into endothelial cells is apparently facilitated by specific membrane receptors and transport systems [16,17,18], which increase the formation of NO through different pathways (some involving ROS) [19,20]. The activation of soluble guanylyl cyclase by NO results in the formation of cyclic guanosine monophosphate (cGMP) that initiate the relaxation of endothelium [12]. However, further research is needed to establish the mechanisms involved in the interaction between the molecules, the triggering of signals and the biological responses. These mechanisms have been studied but mainly with supplements or extracts containing polyphenols at a far higher concentration than those consumed in the usual diet. Moreover, many studies have been performed in elderly populations at risk of developing NCDs due to altered biochemical parameters (high blood pressure, high cholesterol, metabolic syndrome, etc.) or even in patients with a diagnosed disease [21].

Thus, in the present study we aimed to study if a short-term dietary intervention involving a reduced consumption of polyphenol-rich food could affect vascular biomarkers in healthy young men at no apparent risk of developing a disease.

## 2. Materials and Methods

### 2.1. Study Subjects

Twenty-two healthy men between 18 and 32 years of age took part in the interventional study from November 2015 to April 2016. Only male adults were enrolled in this trial to avoid effects related to hormonal fluctuations during the menstrual cycle [22,23]. Exclusion criteria were body mass index (BMI) over 30 kg/m^2^, a history of CVDs, hypertension and dyslipidemia, chronic illness or homeostatic disorder and toxic habits (such as tobacco, alcohol and drug consumption). Volunteers were asked to maintain the same level of physical activity throughout the study. Figure 1 shows the flowchart of subjects in the trial.

All participants provided written informed consent prior to starting the intervention. The study was carried out according to the principles of the Declaration of Helsinki and the protocol was approved by the Ethics Committee of Clinical Investigation of the University of Barcelona (Barcelona, Spain). The clinical trial was registered and given the International Standard Randomized Controlled Trial Number (http://www.isrctn.com/) of ISRCTN17867378.

### 2.2. Study Design

The study was open, crossover and controlled (Figure 2), and every participant underwent two 2-week interventions, and ate their usual diet (UD) and a low antioxidant diet (LAD), with a week of washout in between. Advice about changes in diet was given. Participants following the LAD were asked not to consume more than 2 servings of fruits or vegetables per day and their consumption of polyphenol-rich items, such as coffee, tea or chocolate, was restricted. Appendix A shows all the recommendations to follow a low antioxidant diet (LAD).

In contrast, in the UD there was no restriction on polyphenol-rich food (fruit, vegetables, extra-virgin olive oil, tea, etc.) consumption. The study was run in the Department of Nutrition, Food Science and Gastronomy of the Food Science Torribera Campus of the University of Barcelona (Spain).

### 2.3. Dietary and Physical Activity Assessments

After each intervention, subjects filled out a 3-day food recall questionnaire. The dietary registers were analyzed using nutrition analysis software developed at the University of Barcelona, PCN Pro software (Programa de Càlcul Nutricional Professional, Santa Coloma de Gramenet, Barcelona, Spain). In addition, a 14-point Mediterranean Diet Adherence questionnaire was used to compare the two diets [24]. Physical activity was measured using the validated Spanish version of the Minnesota leisure-time physical activity questionnaire [25].

### 2.4. Blood and Urine Collection

Fasting blood and urine samples were collected after each dietary intervention between 8:00 and 9:00 a.m. Blood was drawn from the arm via venipuncture into tubes containing ethylenediaminetetraacetic acid (EDTA), and plasma was separated after centrifugation at 2070× *g* for 15 min at 4 °C. Plasma and urine were aliquoted and stored at −80 °C until the day of the analysis.

### 2.5. Clinical and Anthropometric Measurements

Diastolic and systolic blood pressure (DBP and SBP) as well as heart rate (HR) were measured in triplicate after each intervention the morning in fasting conditions. Biochemistry parameters in plasma were evaluated at an external laboratory (mdb lab Durán Bellido). C-reactive protein (CRP) was measured by an immunoturbidimetric method. HDL, LDL, total cholesterol and triglycerides were analyzed by an enzymatic method. Urea and uric acid were analyzed by enzymatic and enzymatic/chromogen methods, respectively. Creatinine was determined by reaction kinetics of the Jaffe method (as modified by Larsen). Total proteins and albumin were measured by the endpoint biuret reaction and bromocresol green methods, respectively. Biochemistry measurements were performed once the studied completed in the samples stored at −80 °C.

Body mass index (BMI) was calculated from the weight and height, and the waist-hip ratio (WHR) from the measurements of the waist and hip circumferences taken at the visit after each intervention in the early morning.

### 2.6. Quantification of Total Polyphenol Excretion (TPE) in Urine Samples

A solid phase extraction (SPE) using a 96-well plate cartridge (Oasis MAX, Waters Co., Milford, MA, USA) was performed in diluted urine samples and total polyphenol excretion (TPE) was measured by a Folin–Ciocalteu reaction according to Medina-Remon et al. [26]. The spectrophotometry analysis was carried out at a wavelength of 765 nm. The results were expressed as mg of gallic acid equivalent (GAE)/g of creatinine.

### 2.7. Determination of Plasmatic Inflammatory Biomarkers

The cell adhesion molecules, sICAM-1 and sVCAM-1, were measured by a ProcartaPlex Multiplex Immunoassay kit (Invitrogen, ThermoFisher Scientific, Waltham, MA, USA). Plasma samples were diluted 1:200 and assayed in duplicate. The assay was performed through a MAGPIX^®^ instrument (Luminex, Co., Austin, TX, USA) and data were processed with ProcartaPlex Analyst software.

Before the determination of NO, plasma samples were filtered using the Amicon^®^ Ultra 30K (Merck KGaA, Darmstadt, Germany) device over microcentrifuge tubes and centrifuged at 14,000× *g* for 1 h at 4 °C. The NO amount was indirectly determined using a nitrate/nitrite colorimetric assay kit (Cayman Chem. Co., Ann Arbor, MI, USA, ref. 780001). The detection limit of the assay was approximately 1 µM nitrite, considering a 1:2 dilution of plasma samples. The analysis was carried out in triplicate.

### 2.8. Determination of Eicosanoids in Urine

The PGI_2_ and TXA_2_ were indirectly quantified by measuring PGIM (Prostaglandin I Metabolite) and 11-dehydro thromboxane B_2_, respectively. Both molecules were determined in urine using two competitive enzyme-linked immunosorbent assay (ELISA) kits acquired from Cayman Chem. Co. (Ann Arbor, MI, USA, ref. 501100 and 519510). The PGIM assay has a range from 39 to 5000 pg/mL and a sensitivity (80% B/B0) of approximately 120 pg/mL. The 11-dehydro thromboxane B_2_ assay has a range from 15.6 to 2000 pg/mL and a sensitivity (80% B/B0) of approximately 34 pg/mL. The analysis was carried out in triplicate. The TXA_2_:PGI_2_ ratio was calculated.

### 2.9. Statistical Analysis

Normality was checked by a Shapiro–Wilk test. Non-normal variables were log-transformed. The non-normal variables with values close to zero were transformed by inverse hyperbolic sine (IHS) function. Linear regression models were assayed both to calculate the raw *p*-value and *p*-value of the differences adjusted by mean-centered age (years) and physical activity (METS/day). A *p*-value of <0.05 was considered statistically significant. 

Correlations between the MedDiet score and the analyzed biomarkers were determined using a correlation matrix that considers repeated measures [27]. The Pearson coefficient (*r*) was calculated. Significant correlations (*p* < 0.05) are shown in the results.

All the analysis was performed using the software R, version 3.4.2. (R: A Language and Environment for Statistical Computing. R Foundation for Statistical Computing, Vienna, Austria [28]).

## 3. Results

### 3.1. Characteristics of Participants

Twenty-two men completed the study. Baseline characteristics remained constant throughout the study (age and METS/day). In addition, anthropometric and clinical measurements did not change significantly after the LAD compared to the control (UD) (*p* > 0.05) (Table 1).

### 3.2. Dietary Intake

Table 2 shows the differences between the two interventions (UD and LAD) in nutrients and healthy food intake. Although the diets were similar in macronutrients (except for the fat content), the LAD had a lower concentration of vitamins, minerals, and fiber. Levels of magnesium, potassium and iron differed significantly between the diets, as did vitamins A, E, C and folate (*p* ≤ 0.001), because the UD involved a higher consumption of fruits and vegetables rich in these compounds (*p* < 0.001), as well as of cocoa (*p* < 0.001), tea (*p* < 0.001), olive oil (*p* = 0.01), and nuts and seeds (*p* < 0.001).

The score categories of adherence to the MedDiet are high (≥10), moderate (6–9) and low (≤5). According to the scores obtained in the 14-item questionnaire, the adherence to the MedDiet in the UD was moderate (8.5 ± 0.4), and low in the LAD (4.7 ± 0.3), the two dietary interventions being significantly different (*p* < 0.001).

### 3.3. TPE in Urine after Each Dietary Intervention

Figure 3 shows a significant decrease of TPE, expressed in gallic acid equivalents (GAE) and corrected by creatinine, in participants after following the LAD for two weeks. The difference in TPE between the two interventions was around 45 mg GAE/g creatinine (*p* = 0.007).

### 3.4. Inflammatory Molecules in Plasma and Urine

Significant changes in NO levels (*p* = 0.001 and adjusted *p* = 0.002) were observed. Figure 4A shows the reduction of NO concentration after the short-term diet with food containing lower amounts of bioactive compounds compared to the UD intervention (52 ± 28 µM in LAD vs. 80 ± 34 µM in UD).

Although there were negative changes in the inflammatory parameters (cellular adhesion molecules and eicosanoids) after the LAD, these differences were not significant (Figure 4B–E). The TXA_2_:PGI_2_ ratio, an indicator of the balance of platelet function, increased considerably from 0.09 ± 0.08 to 0.16 ± 0.14 (*p* = 0.051 and adjusted *p* = 0.048, Figure 4F). 

### 3.5. Correlation between Biomarkers and Mediterranean Diet Adherence

After the short-term intervention with low-antioxidant foods, significant changes were observed in the TPE and NO. Both parameters correlated positively with the MedDiet. Thus, TPE in urine and NO concentration in plasma had decreased in participants after following the diet poor in bioactive compounds (Figure 5A,B).

## 4. Discussion

In our study, changing to a diet low in polyphenols affected NO expression and modified the anti-inflammatory response in young healthy men, although without reaching significance, except for the ratio TXA_2_:PGI_2_ probably due to the low number of subjects and the short period of the intervention.

Regarding the results of the health-related biomarkers, our study suggests that following a diet low in bioactive compounds, measured by the validated biomarker of TPE, for a period as short as two weeks, reduced plasmatic NO in healthy volunteers. Several studies have shown the protective role of polyphenols or polyphenol-rich food on endothelium function [10] and that following a Mediterranean diet is beneficial for vascular health. Schroeter et al. reported an acute increase in the concentration of circulating NO species after the consumption of flavanol-rich cocoa in a crossover trial with healthy male subjects [29]. These authors also observed a positive correlation between the chronic ingestion of flavanol-rich cocoa and the urinary excretion of NO metabolites, at least partially due to the flavanol (−)-epicatechin, in a cross-sectional study of the Kuna population [29]. Likewise, in elderly participants at high cardiovascular risk from the PREDIMED trial, a significant increase of total plasma NO was observed after a one-year intervention with an extra intake of extra virgin olive oil or nuts [8]. This increase was associated with a significant change in the blood pressure (BP) and correlated with the TPE in urine [8]. In another PREDIMED substudy, an increase in serum NO (nitrate + nitrite) after a Mediterranean Diet + extra virgin olive oil intervention was negatively correlated with BP in hypertensive women [30]. In addition, in a crossover study where healthy subjects consumed an ice cream based on dark cocoa powder with a hazelnut and green tea extract, an increase in total serum polyphenol and NO levels were found and correlated [31]. 

Although in our study a significant change in BP and higher NO levels did not correlate with TPE in urine, a positive correlation was obtained between total plasma NO and MedDiet adherence. Thus, NO levels in plasma were lower in participants after following the LAD rather than the UD (*p* = 0.008, *r* = 0.54). Other authors performing crossover trials where healthy subjects consumed polyphenol-rich products or extracts have also found an increase of plasma or urinary NO [32,33,34]. In a parallel study with elderly pre-hypertensive or stage 1 hypertensive participants without concomitant risk factors, significant changes in BP and NO were observed after an intervention with dark chocolate for 18 weeks [35]. A reduction in BP and a rise of plasma NO, both correlated with each other, were observed in men at high cardiovascular risk after an intervention with dealcoholized red wine in a crossover trial [36]. In addition, 24 men with metabolic syndrome consuming a grape supplement in a 30-day-crossover trial had higher levels of plasma NO, correlated with a decrease in systolic BP, although the change in NO was not significant [21]. However, in contrast with the aforementioned studies, a recent crossover trial studying the acute effect of coffee in healthy volunteers did not find significant differences in plasma NO concentrations or BP [37]. Likewise, no changes in concentration of plasma NO were observed in subjects at risk of CVDs consuming a beverage based on wild blueberries daily for six weeks [38].

The pro-inflammatory response increased in participants after the two-week LAD, although changes in plasma ICAM-1 nor VCAM-1 were not significant. In accordance with our results, Taubert et al. reported lower levels of ICAM-1 in plasma after the consumption of a grape supplement, and although there was not a significant difference in VCAM-1, the reduction of this biomarker was positively correlated with a decrease in systolic BP [35]. Similar results were obtained in another parallel trial where hypertensive subjects consumed 150 mL/day of pomegranate juice for 2 weeks and a reduction of the VCAM-1 biomarker and BP was observed [39]. Several other authors have observed positive effects on CAMs after consumption of polyphenol extracts or polyphenol-rich products [6,9,40,41]. However, another trial did not find any effect on ICAM-1 or VCAM-1 [38,42,43]. 

The balance of the vasodilation and vasoconstrictor molecules, PGI_2_ and TXA_2_, is also crucial for vascular health. Both eicosanoids are synthesized from arachidonic acid by cyclooxygenase isoforms (COX_1_ and COX_2_). In the present study, a notable increase in the TXA2:PGI2 ratio was observed, being almost double after the LAD intervention compared to the UD (*p* = 0.048), suggesting that polyphenols could contribute to the maintenance of vascular homeostasis. The observed trend was for PGI2 to decrease and TXA2 to increase after the LAD, although the change of both biomarkers did not reach significance. Dietary polyphenols regulate immune cells as well as the expression of several pro- and anti-inflammatory genes and cytokines by different pathways (MAPK, NF-kB and arachidonic acid), contributing to anti-inflammatory, immune-modulatory, and antioxidant activities [44]. 

The effect of polyphenol-rich products on eicosanoids has been demonstrated. Pignatelli et al. showed that following a Mediterranean diet reduces the level of TXA_2_ in atrial fibrillation patients [45]. Also, the intake of extra virgin olive oil, a typical polyphenol-rich product of the Mediterranean diet, reduced the serum TXB_2_ concentration in healthy subjects [46]. However, other authors have obtained inconclusive results. In a crossover study, platelet TXA_2_ production in smokers, which was higher than in healthy subjects, did not change after consuming 40 g of milk chocolate or of dark chocolate [47]. No effect on TXA2 and PGI2 metabolites in urine, or on the ratio of both molecules, was found in healthy subjects consuming an American diet and another supplemented with cacao/chocolate (containing 466mg of procyanidins) for four weeks [48]. Moreover, after consuming a serving of aronia-citrus juice for 45 days, 16 triathletes had lower urinary excretion of 11-dehydro-TXB_2_, but no significant change was observed in the 2,3-dinor-6-keto-PGF1α (PGI_2_ metabolite) [49].

A strong point of this study is its focus on healthy young volunteers (most of them university students), which is atypical in this research area. Moreover, the intervention was carried out with a normal diet, reducing the intake of polyphenol-rich food for only a short period. The limitations of the study design are the small size of the sample, limited to a male population, the non-randomized design, and the lack of baseline measures. However, as the UD was the normal diet of the volunteers, it was not considered necessary to take baseline measures into account (data not shown). 

## 5. Conclusions

These results highlight that adopting a diet low in polyphenols for a period as short as two weeks can alter the total plasma NO concentration in healthy young men. An increasing imbalance between TXA_2_ and PGI_2_ was also observed, although the changes in both eicosanoids did not reach significance. Thus, this study confirms the health protective role of polyphenol-rich food and suggests that avoiding or reducing the consumption of food rich in these compounds has a negative effect on vascular biomarkers, even after a short period and in healthy subjects.

## Figures and Tables

**Figure 1 nutrients-10-01766-f001:**
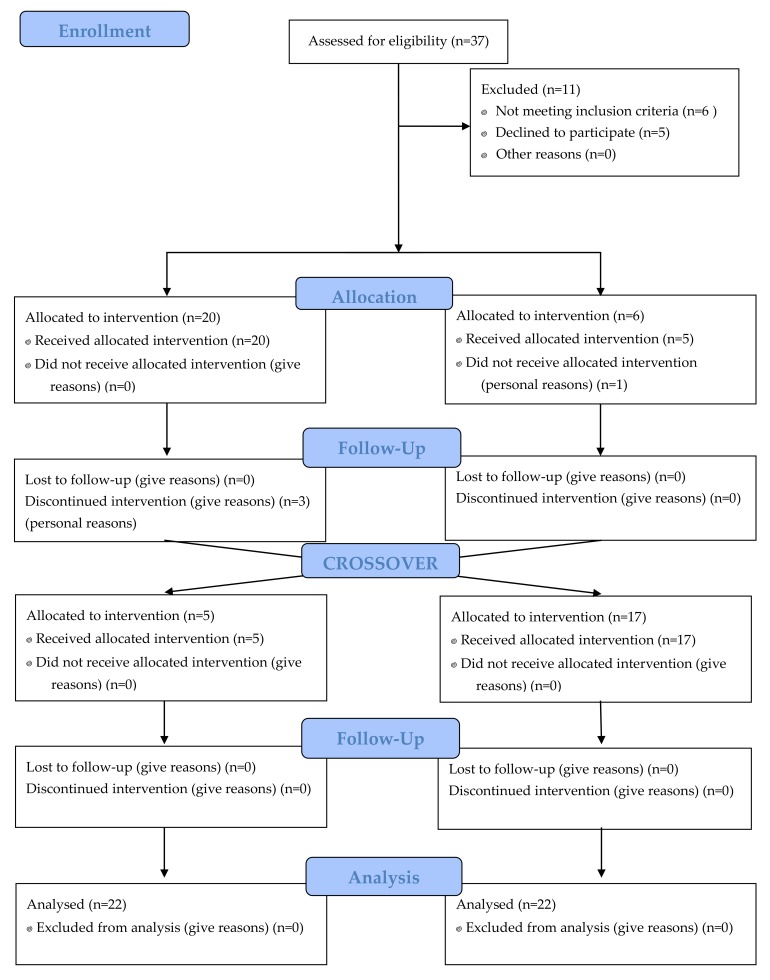
Flow diagram showing the participation of volunteers in each phase of the trial. The usual diet (UD) is shown on the left side of the graph and the low antioxidant diet (LAD) on the right side.

**Figure 2 nutrients-10-01766-f002:**
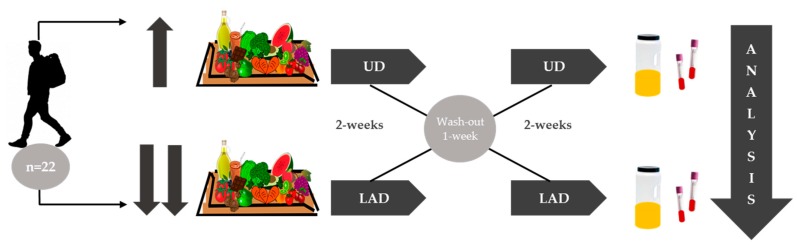
General scheme of the crossover trial with two intervention periods.

**Figure 3 nutrients-10-01766-f003:**
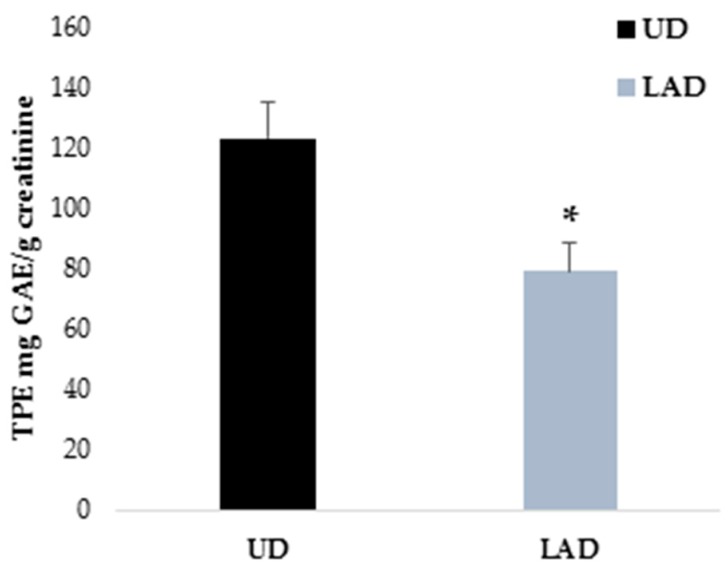
Comparison of total polyphenol excretion (TPE) in urine after each intervention. Data shown are mean ± SEM. * Significant differences with *p* < 0.05.

**Figure 4 nutrients-10-01766-f004:**
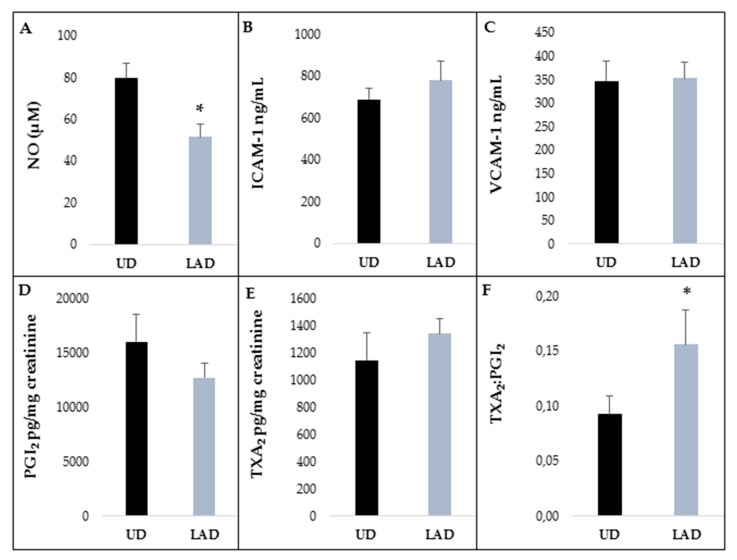
(**A**) NO (**B**) ICAM-1 (**C**) VCAM-1 (**D**) PGI_2_ (**E**) TXA_2_ and (**F**) ratio TXA_2_/PGI_2_ between both interventions. Data shown are mean ± SEM. The UD is in black and the LAD is in gray; * significant differences with *p* < 0.05.

**Figure 5 nutrients-10-01766-f005:**
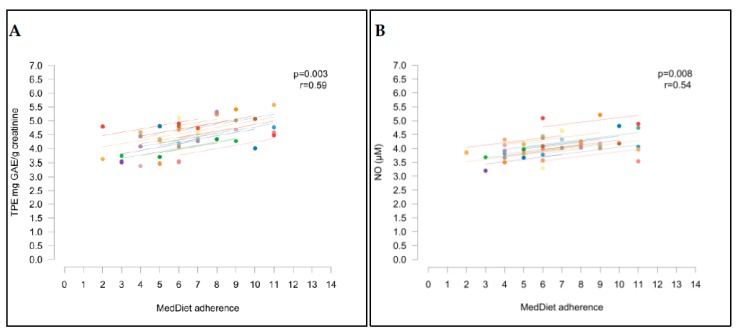
Correlation of Mediterranean diet adherence with TPE (**A**) and NO (**B**).

**Table 1 nutrients-10-01766-t001:** Characteristics of all participants of the study.

	UD	LAD	*p*	Adjusted *p* ^a^
Baseline characteristics				
Age (years)	24 ± 1			
Physical activity (METS/day)	743 ± 75			
Anthropometric measurements				
BMI (kg/m^2^)	24.9 ± 0.79	24.8 ± 0.79	0.375	0.33
WHR	0.84 ± 0.01	0.84 ± 0.01	1	0.89
Clinical measurements				
SBP (mmHg)	123 ± 2	122 ± 2	0.864	0.90
DBP (mmHg)	76 ± 2	76 ± 2	0.893	0.995
HR (bpm)	66 ± 2	63 ± 2	0.109	0.198
CRP (mg/dL)	0.10 ± 0.02	0.09 ± 0.01	0.893	0.925
Total cholesterol (mmoles/L)	3.84 ± 0.12	3.80 ± 0.12	0.547	0.551
HDL (mmoles/L)	1.37 ± 0.06	1.36 ± 0.05	0.865	0.832
LDL (mmoles/L)	2.04 ± 0.11	1.97 ± 0.11	0.250	0.306
Triglycerides (mmoles/L)	0.94 ± 0.08	1.04 ± 0.12	0.470	0.525
Urea (mmoles/L)	5.56 ± 0.30	5.94 ± 0.25	0.211	0.365
Creatinine (µmoles/L)	76 ± 1.51	76.3 ± 1.60	0.510	0.752
Uric acid (µmoles/L)	319 ± 10.95	332 ± 10.51	0.138	0.136
Total proteins (g/L)	73.1 ± 0.57	73.9 ± 0.54	0.272	0.324
Albumin (g/L)	46.8 ± 0.50	46.6 ± 0.41	0.655	0.608

Data are mean ± standard error of the mean (SEM). ^a^ Baseline measures, age and physical activity were used to adjust the *p*-value.

**Table 2 nutrients-10-01766-t002:** Changes in the dietary parameters after 2 weeks.

	UD	LAD	*p*	Adjusted *p* ^a^
Nutrients				
Energy (kcal/day)	2393 ± 130	2194 ± 127	0.083	0.033 *
Carbohydrates (g/day)	256 ± 15	245 ± 14	0.445	0.498
Fat (g/day)	106 ± 9	83 ± 7	0.005 *	0.001 *
Protein (g/day)	100 ± 6	109 ± 7	0.256	0.473
Cholesterol (mg/day)	301 ± 30	241 ± 21	0.046 *	0.025 *
Fiber (g/day)	30 ± 2	16 ± 1	<0.001 *	<0.001 *
Ca (mg/day)	869±49	936 ± 91	0.452	0.816
P (mg/day)	1599±82	1612 ± 97	0.897	0.711
Mg (mg/day)	436±25	314 ± 30	<0.001 *	<0.001 *
Na (mg/day)	2495±273	2945 ± 280	0.065	0.107
K (mg/day)	4237±180	2950 ± 325	<0.001 *	<0.001 *
Fe (mg/day)	17±0.98	12 ± 0.99	0.001 *	0.002 *
Zn (mg/day)	11±0.72	10 ± 0.60	0.217	0.057
Vit. A (mcg r.e./day)	1460±232	242 ± 45	<0.001 *	<0.001 *
Vit E (mg t.e./day)	15±1.19	8 ± 0.70	<0.001 *	<0.001 *
Vit. C (mg/day)	255±25	39 ± 6	<0.001 *	<0.001 *
Folate (mcg/day)	511±34	243 ± 33	<0.001 *	<0.001 *
Food				
Vegetable (g/day)	402 ± 38	120 ± 25	<0.001 *	<0.001 *
Fruit (g/day)	368 ± 53	48 ± 16	<0.001 *	<0.001 *
Nuts and seeds (g/day)	15 ± 5	0.2 ± 0.2	<0.001 *	<0.001 *
Virgin olive oil (g/day)	33 ± 2	26 ± 2	0.009 *	0.01 *
Pulses (g/day)	19 ± 6	16 ± 6	0.210	0.295
Cereals (g/day)	202 ± 20	275 ± 17	0.001 *	0.001 *
Cocoa (g/day)	13 ± 3	0.7 ± 0.6	<0.001 *	<0.001 *
Coffee (g/day)	63 ± 24	23 ± 9	0.186	0.162
Tea (g/day)	112 ± 29	0	<0.001 *	<0.001 *
White meat (g/day)	70 ± 14	79 ± 14	0.353	0.312
Fish (g/day)	61 ± 12	65 ± 17	0.768	0.885
Wine (g/day)	5 ± 3	0	0.163	0.107
Beer (g/day)	16 ± 8	13 ± 10	0.201	0.215
Mediterranean diet score				
MedDiet score ^†^	8.5 ± 0.4	4.7 ± 0.3	<0.001 *	<0.001 *

Data are mean ± SEM. ^†^ MedDiet score obtained from 14-item Mediterranean Diet Adherence Screener. ^a^ Baseline measures, age and physical activity were used to adjust the *p*-value. * Significant difference (*p* < 0.05).

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
