# Peer review of "Changing to a Low-Polyphenol Diet Alters Vascular Biomarkers in Healthy Men after Only Two Weeks"

_nutrients, 2018, doi:10.3390/nu10111766_

Round 1
Reviewer 1 Report
This an interesting and well-written manuscript, which aims to study whether a short-term dietary intervention involving a reduced consumption of polyphenol-rich food could affect vascular biomarkers in healthy young men.
In my opinion, the introduction provides sufficient background and references.
The research design is appropriate, though sample size is limited. Moreover, my major concern regards the lack of baseline measures that Authors have addressed in the text.
Did the Authors check for normality? If distribution of some variables was skewed, I suggest to use non parametric test or log-trasformation
Methods are adequately described. In this section, I suggest to include more details in the statistically analysis paragraph, especially on the test used for unadjusted comparisons.
Results are clearly presented, but I suggest to include p-value for all significant comparisons.
Conclusion are supported by the results. Recently, polyphenols (e.g. resveratrol) have been demonstrated to modulate epigenetic mechanisms in cell and tissue under oxidative stress (please consider as an example doi: 10.3390/ijms19072118). I propose to the Authors to include some lines on this novel evidence.
Author Response
Comments and Suggestions for Authors
This an interesting and well-written manuscript, which aims to study whether a short-term dietary intervention involving a reduced consumption of polyphenol-rich food could affect vascular biomarkers in healthy young men.
In my opinion, the introduction provides sufficient background and references.
C.1. The research design is appropriate, though sample size is limited. Moreover, my major concern regards the lack of baseline measures that Authors have addressed in the text.
R.1. We agree with the reviewer and for this reason we included that as a limitation in the discussion of the manuscript. However, we think that it does not affect the results of our study because we checked the habitual diet of the participants during the screening visit and we observed that there were not differences between their regular diets (which would be the baseline diet) and the usual diets (the intervention diet) using the 14-item Mediterranean diet assessment (data not shown). We, thus, verified that usual diet records were consistent with the screening visit records.
This clarification has been added to the discussion (lines 298-301).
C.2. Did the Authors check for normality? If distribution of some variables was skewed, I suggest to use non parametric test or log-trasformation
R.2. We have reviewed and corrected the statistical analysis according with your comments. We have log-transformed the non-normal variables. In some variables log-transformations were not possible because the values were close to the zero (some parameters of the diet like the intake of wine), then we normalized those variables by an inverse hyperbolic sine (IHS) transformation. We modifeied the results of the manuscript according with the corrected statistical treatment, even the p-values changed the results were consistent with the previous ones and they did not changed except for the ratio TXA2/PGI2 parameter that is now significative.
C.3. Methods are adequately described. In this section, I suggest to include more details in the statistically analysis paragraph, especially on the test used for unadjusted comparisons.
R.3. We detailed the section 2.9. Statistical analysis. We have included the details about the normality test used and the transformation of non-normal variables performed (lines 150-152). Moreover, we clarified in the lines 152-153 of the manuscript that linear regression models were not only used to obtain the adjusted p-value but also the unadjusted p value. In addition, we added the link about the package used to repeated measures correlation (lines 156-157).
C.4. Results are clearly presented, but I suggest to include p-value for all significant comparisons.
R.4. Thanks for your observation, we included all the p-values missing.
C.5. Conclusion are supported by the results. Recently, polyphenols (e.g. resveratrol) have been demonstrated to modulate epigenetic mechanisms in cell and tissue under oxidative stress (please consider as an example doi: 10.3390/ijms19072118). I propose to the Authors to include some lines on this novel evidence.
R.5. We considered the reviewer’s proposal and included in the discussion a paragraph describing this evidence (lines 279-282). We cited a recent published about this content (reference 42).

Reviewer 2 Report
The article "Changing to a low polyphenol diet alters vascular biomarkers in healthy men after only two weeks" discusses the short term effects of a polyphenol-rich and polyphenol-poor diet in healthy young adult men. The study is novel in that it examines a healthy population and demonstrates changes that occur after a short-period of adding in polyphenols.
The introduction is well written, however, the last paragraph should be expanded upon:
A citation should be provided for line 58-59: "which increase the formation of NO through different pathways (some involving ROS)".
The following line, 59-60, seems to discuss guanylyl cyclase and cGMP too briefly without describing it's purpose.
Methods:
How were participants randomized? Why were 20 randomized to the UD group and 6 to the LAD group first?
In subsection 2.5 Clinical and anthropometric measurements, when were these measures taken? Please include the time points in this section.
Results:
Throughout results and tables, the description of p value needs to be uniform. Sometimes it is written "p value < 0.05" and sometimes written "p<0.05". I think the word "value" can be dropped, however, it just needs to be uniform.
Line 164, "p=" needs to be added to tea, olive oil, and nuts and seeds. Currently there are only unidentified numerical values.
Table 1:
Heading "LD" should be "LAD".
Table 2:
Significant values need asterisk: Nuts and seeds, virgin olive oil, tea.
Figure 4 and Table 3:
Seem redundant. What is the difference between these two data sets?
Discussion
The lack of baseline measures was briefly mentioned, however, it was rationalized that it wasn't needed due to the UD being a normal diet. However, baseline measures are needed to address the following questions: Was the 1 week washout period sufficient to bring participants back to baseline? Did participants consume more polyphenols in there usual diet that normal due to being in the study?
Author Response
REVIEWER 2
Comments and Suggestions for Authors
The article "Changing to a low polyphenol diet alters vascular biomarkers in healthy men after only two weeks" discusses the short term effects of a polyphenol-rich and polyphenol-poor diet in healthy young adult men. The study is novel in that it examines a healthy population and demonstrates changes that occur after a short-period of adding in polyphenols.
The introduction is well written; however, the last paragraph should be expanded upon:
C.1. A citation should be provided for line 58-59: "which increase the formation of NO through different pathways (some involving ROS)".
R.1.The references “19” and “20” have been added supporting this data.
C.2. The following line, 59-60, seems to discuss guanylyl cyclase and cGMP too briefly without describing it's purpose.
R.2. This sentence has been completed clarifying the role of cGMP produced after the activation of guanylyl cyclase by NO (lines 59-61).
Methods:
C.3. How were participants randomized? Why were 20 randomized to the UD group and 6 to the LAD group first?
R.3. Thanks for the reviewer comment, we revised and modified the manuscript. It was an error to consider the study randomized because due to logistic reason we could only randomized half of the volunteers, that is why 20 started UD and 6 LAD. We removed “randomized” in the point 2.2. Study design, line 91 of the manuscript and corrected the flowchart to non-randomized trial design. Also, we consider this point between the limitations of the study in the discussion (lines 298-301).
C.4. In subsection 2.5 Clinical and anthropometric measurements, when were these measures taken? Please include the time points in this section.
R.4. According your indication, we clarified the time points in this section of manuscript (lines 115, 121-122 and 124-125 of the section 2.5. Clinical and anthropometric measurements).
Results:
C.5. Throughout results and tables, the description of p value needs to be uniform. Sometimes it is written "p value < 0.05" and sometimes written "p<0.05". I think the word "value" can be dropped, however, it just needs to be uniform.
R.5. We have changed “p value” to “p” in the results and tables to be uniform.
C.6. Line 164, "p=" needs to be added to tea, olive oil, and nuts and seeds. Currently there are only unidentified numerical values.
R.6. We corrected these mistakes in the manuscript.
C.7. Table 1:
Heading "LD" should be "LAD".
R.7. We corrected this mistake.
C.8. Table 2:
Significant values need asterisk: Nuts and seeds, virgin olive oil, tea.
R.8. We inserted asterisk in the significant values.
C.9. Figure 4 and Table 3:
Seem redundant. What is the difference between these two data sets?
R.9. As you pointed out, we removed the “Table 3” of the manuscript to avoid the redundant information.
C.10. Discussion
The lack of baseline measures was briefly mentioned, however, it was rationalized that it was not needed due to the UD being a normal diet. However, baseline measures are needed to address the following questions: Was the 1 week1-week washout period sufficient to bring participants back to baseline? Did participants consume more polyphenols in there usual diet that normal due to being in the study?
R.10. We did not believe the necessity of baseline because we checked that usual diet was like their regular diet (data not shown). On the one hand, we think that 1-week washout period was enough taking account that each intervention lasts two weeks and considered usual diet like a basal diet. And on the other hand, we showed the score from 14-item Mediterranean Diet questionnaire and we confirmed that volunteers kept their dietetic habits. Intermediate adherence to Mediterranean diet was observed in the usual diet.
This clarification has been added to the discussion (lines 298-301).

Reviewer 3 Report
Hurtado-Barroso et al. in the present cross-over study “Changing to a low-polyphenol diet alter vascular biomarkers in healthy men after only two weeks” investigated the effects of diet low in polyphenol-rich food on biomarkers of endothelial dysfunction in healthy male individuals. Authors observed significantly less urine polyphenols in the Low antioxidants diet (LAD) group compared to the usual diet (UD) group. They also found reduced plasma NO levels in the LAD group, however, no significant difference was observed in VCAM, ICAM and prostacyclin levels between two groups. Authors concluded that decreased intake of polyphenolic food can affect endothelial function and plasma NO levels in healthy individuals. The manuscript is well written and data are clearly presented. However, I have a few comments:
1. Authors did not investigate levels of all above-mentioned parameters prior to the start of the study. The comparison of baseline and end-point parameters would have provided the true variation in these parameters following the change in dietary habits.
2. They observed significantly reduced plasma NO levels in individuals with LAD. However, it is not clear whether this decrease in NO levels is due to impaired NO production or increased reaction of NO with superoxide, and subsequent peroxynitrite formation. Authors should have investigated plasma levels of superoxide dismutase or peroxynitrite in various groups.
3. Conclusions of the study are not supported by the presented data. They did not perform any non-invasive endothelial function test like branchial flow-mediated dilation on these subjects. They could have analyzed more markers of endothelial dysfunction such as CRP, angiotensin II, PAI1 and ADMA etc. Furthermore, CRP has been shown to modulate NO levels.
Author Response
REVIEWER 3
Comments and Suggestions for Authors
Hurtado-Barroso et al. in the present cross-over study “Changing to a low-polyphenol diet alter vascular biomarkers in healthy men after only two weeks” investigated the effects of diet low in polyphenol-rich food on biomarkers of endothelial dysfunction in healthy male individuals. Authors observed significantly less urine polyphenols in the Low antioxidants diet (LAD) group compared to the usual diet (UD) group. They also found reduced plasma NO levels in the LAD group, however, no significant difference was observed in VCAM, ICAM and prostacyclin levels between two groups. Authors concluded that decreased intake of polyphenolic food can affect endothelial function and plasma NO levels in healthy individuals. The manuscript is well written and data are clearly presented. However, I have a few comments:
C. 1. Authors did not investigate levels of all above-mentioned parameters prior to the start of the study. The comparison of baseline and end-point parameters would have provided the true variation in these parameters following the change in dietary habits.
R.1. We agree with the reviewer and for this reason we included that as a limitation in the discussion of the manuscript. However, we think that it does not affect the results of our study because we checked the habitual diet of the participants during the screening visit and we observed that there were not differences between their regular diets (which would be the baseline diet) and the usual diets (the intervention diet) using the 14-item Mediterranean diet assessment (data not shown). We, thus, verified that usual diet records were consistent with the screening visit records.
This clarification has been added to the discussion (lines 298-301).
C. 2. They observed significantly reduced plasma NO levels in individuals with LAD. However, it is not clear whether this decrease in NO levels is due to impaired NO production or increased reaction of NO with superoxide, and subsequent peroxynitrite formation. Authors should have investigated plasma levels of superoxide dismutase or peroxynitrite in various groups.
R.2. Your proposal seems very interesting to us, however we did not consider to evaluate SOD and peroxynitrite in the study design. Since we rely on the methodology described in some studies that investigated this parameter (Chiva-Blanch et al. Dealcoholized Red Wine Decreases Systolic and Diastolic Blood Pressure and Increases Plasma Nitric Oxide: Short Communication. Circ. Res. 2012, 111, 1065–1068 and Medina-Remón et al. PREDIMED Study Investigators Effects of total dietary polyphenols on plasma nitric oxide and blood pressure in a high cardiovascular risk cohort. The PREDIMED randomized trial. Nutr. Metab. Cardiovasc. Dis. 2015, 25, 60–7). Moreover, we had the problem that we had a not enough volume of biological sample to perform this analysis.
C. 3. Conclusions of the study are not supported by the presented data. They did not perform any non-invasive endothelial function test like branchial flow-mediated dilation on these subjects. They could have analyzed more markers of endothelial dysfunction such as CRP, angiotensin II, PAI1 and ADMA etc. Furthermore, CRP has been shown to modulate NO levels.
R.3. We agree with your comment about the conclusions and modified this section in the manuscript. Moreover, we have included the marker CRP (lines 116-117, section 2.5. Clinical and anthropometric measurements and Table 1) that was not find necessary because there were no differences between the two interventions. We thank your suggestion about the evaluation of complementary markers and will consider these parameters in future studies. We had a limited biological sample volume to carry out the assessments.
